# A Novel Non-Metallic Photocatalyst: Phosphorus-Doped Sulfur Quantum Dots

**DOI:** 10.3390/molecules28083637

**Published:** 2023-04-21

**Authors:** Ziyi Liu, Chuanfu Shan, Guiyu Wei, Jianfeng Wen, Li Jiang, Guanghui Hu, Zhijie Fang, Tao Tang, Ming Li

**Affiliations:** 1College of Science & Key Laboratory of Low-Dimensional Structural Physics and Application, Education Department of Guangxi Zhuang Autonomous Region, Guilin University of Technology, Guilin 541004, China; 2School of Electronics Engineering, Guangxi University of Science and Technology, Liuzhou 545006, China

**Keywords:** phosphorus-doping, sulfur quantum dots, photocatalysis, charge transfer

## Abstract

In this paper, a novel phosphorus-doped sulfur quantum dots (P-SQDs) material was prepared using a simple hydrothermal method. P-SQDs have a narrow particle size distribution as well as an excellent electron transfer rate and optical properties. Compositing P-SQDs with graphitic carbon nitride (g-C_3_N_4_) can be used for photocatalytic degradation of organic dyes under visible light. More active sites, a narrower band gap, and stronger photocurrent are obtained after introducing P-SQDs into g-C_3_N_4_, thus promoting its photocatalytic efficiency by as much as 3.9 times. The excellent photocatalytic activity and reusability of P-SQDs/g-C_3_N_4_ are prospective signs of its photocatalytic application under visible light.

## 1. Introduction

The energy crisis, environmental pollution, and climate change caused by excessive consumption of fossil energy have become the focus of global attention [1]. Currently, human demand for energy is increasing, and fossil energy extraction and use has brought about serious environmental pollution problems. Various organic dyes, chemicals, and other pollutants are discharged directly into the water, causing serious water pollution. Some conventional wastewater treatment methods are not effective in removing these widely available organic pollutants (e.g., dyes, antibiotics, pharmaceuticals). Therefore, how to effectively combat water pollution has become an important issue for researchers [2,3].

Photocatalysis is of great interest to researchers, because it can use natural solar energy as a driving force to achieve photocatalytic decomposition of water for hydrogen production and degradation of organic pollutants [4]. In the last decade, many photocatalysts have been investigated, such as biochar (BC) [5], graphitic phase carbon nitride (g-C_3_N_4_) [6], titanium dioxide (TiO_2_) [7,8], and various other new green and environmentally friendly materials. Among them, g-C_3_N_4_ is widely used because it is superior in easy synthesis, unique electronic band structure and high physicochemical stability [9,10,11,12,13,14]. However, the small specific surface area, narrow visible light absorption range, and fast complexation of photogenerated electron-hole pairs seriously limit its practical application in photocatalysis. Therefore, compositing other materials with g-C_3_N_4_ to improve the photogenerated carrier utilization efficiency has become one of the current research hotspots in the photocatalysis field [15]. Carbon-based quantum dots (CQDs), as one of the most well-known non-metallic elemental nanodots, have been widely used to composite with g-C_3_N_4_ to form heterojunctions to substantially improve the photocatalytic efficiency [16,17,18,19], due to their excellent optical and electrical properties [20]. Interestingly, sulfur quantum dots (SQDs), another type of completely non-metallic nanodots, have attracted less attention in photocatalysis although they have similar optical properties to CQDs.

Elemental sulfur is one of the most abundant substances on Earth and has been widely used for centuries. In the last few years, SQDs, a new class of non-metallic fluorescent nanomaterials, have attracted great interest due to their unique optical, spectroscopic, chemical and antimicrobial properties [21,22]. In 2014, Li et al. demonstrated for the first time that zero-valent sulfur monomers can be prepared as fluorescent quantum dots and pointed out their potential for energy and photocatalytic applications [23]. At present, great progress has been made in the synthesis of SQDs. Shen et al. synthesized SQDs in one step and the prepared SQDs possess good stability and temperature responsiveness, and exhibit strong photocatalytic activity [24]. The SQDs were synthesized by Song et al. and possess advantages such as low cost, surface functionalization convenience, and unique photoluminescence (PL) properties, thus being able to be used as photosensitizers to broaden the light absorption range of g-C_3_N_4_ and improving the photocatalytic activity of g-C_3_N_4_ in visible light [25]. Cheng et al. reported that α-sulfur crystals produce hydroxyl radicals (·OH) under light and have the ability to decompose RhB [26]. However, SQDs based on a single element S lack the adjustment methods to tune the physiochemical properties, thus limiting the further application in the photocatalytic field. Heteroatom doping is an effective way to tune the physical properties of experimental materials. Additionally, among the non-metal doping, sulfur-doped g-C_3_N_4_ can change the electronic structure of g-C_3_N_4_, adjust the position of conduction band (CB) and valence band (VB), enhance the carrier mobility, and improve the photocatalytic activity. Phosphorus (P), a widely used dopant in carbon-based materials, has been proved to successfully modifying the physicochemical properties of GQDs, such as tuning the electronic structure, introducing surface functional groups and enhancing the electron transfer properties, thus significantly enhancing the photodegradation ability of organic dyes when compositing P-GQDs with g-C_3_N_4_ [16]. P-doping is believed to have a similar modifying effect in SQDs.

In this work, phosphorus doping of SQDs and modification of g-C_3_N_4_ were used via a hydrothermal method, and the prepared composites exhibited excellent photocatalytic degradation of colored dyes under visible light irradiation, being the first composite of hetero-element-doped SQDs for photocatalytic applications. This study provides experimental guidance for the design and synthesis of nanostructured photocatalysts in the future by elucidating their visible photocatalytic mechanism while expanding the applications of g-C_3_N_4_-based photocatalysts in the fields of pollutant degradation, solar cells, and artificial photosynthesis for hydrogen production.

## 2. Results and Discussion

### 2.1. Morphological Structure Characterization

The microstructural morphologies of the g-C_3_N_4_ and P-SQDs/g-C_3_N_4_ samples were investigated using TEM and SEM. We found that all these P-SQDs (Figure 1a) exhibit good uniform dispersion and high crystal structure with a spherical shape. The obtained P-SQDs show a uniform size distribution with an average diameter of 3 nm for P-SQDs (Figure 1b). In addition, the lattice spacing of P-SQDs was clearly shown to be 0.46 nm in HRTEM (Figure 1a, inset). The surface morphology and elemental composition of the synthesized photocatalysts were verified through SEM, respectively (Figure 1c–f). The pure g-C_3_N_4_ (Figure 1c) shows an aggregated lamellar structure with an irregularly shaped graphite structure, which is typical of g-C_3_N_4_ synthesized via polymerization methods [27]. SEM images of micron-scale P-SQDs/g-C_3_N_4_ (Figure 1d,e) show that P-SQDs molecules are uniformly and tightly attached to the surface of g-C_3_N_4_. In Figure 1f, the distributions of P, S on P-SQDs/g-C_3_N_4_ in the P-SQDs/g-C_3_N_4_ composite are observed. It clearly shows the good distribution of P and S elements in P-SQDs/g-C_3_N_4_. These results strongly demonstrate the uniform distribution of P-SQDs and the successful composition of g-C_3_N_4_ with P-SQDs. In Figure 1d,e, the cohesive lamellar structure of P-SQDs/g-C_3_N_4_ shows many bends, folds, and overlapping layers, and the unique structure facilitates the migration of charge carriers and the diffusion of substances in an aqueous solution. This would be beneficial as regards improving the activity of photocatalytic reactions.

The specific surface area and pore structure of g-C_3_N_4_ and P-SQDs/g-C_3_N_4_ photocatalysts were investigated via nitrogen adsorption–desorption analysis, and the results are shown in Figure 2. The specific surface area and pore size of P-SQDs/g-C_3_N_4_ composites are superior to those of pure g-C_3_N_4_. In Figure 2a, it can be seen that both g-C_3_N_4_ and P-SQDs/g-C_3_N_4_ are type IV isotherms with hysteresis loops. The increase in specific surface area of P-SQDs/g-C_3_N_4_ is due to the increase in pore volume in the diameter range of 2–30 nm (Figure 2b), corresponding to the accumulation of broken small particles due to partial polymerization [28]. P-SQDs/g-C_3_N_4_ exhibits the largest pore volume in the range of 2–5 nm in diameter [29]. It was demonstrated that by modifying g-C_3_N_4_ with P-SQDs, not only can the specific surface area be increased, but the pore size can also be increased to make them richer in active sites, which is conducive to improving the adsorption capacity of the composites on dye wastewater.

### 2.2. Nanostructure Investigation

XRD is commonly used to determine the crystal structure of the material (Figure 3a). In the XRD pattern of g-C_3_N_4_, two distinct diffraction peaks were observed. The peak at 13.1° indicates that the (100) crystal plane produces tri-S-triazine units, while the other peak at 27.3° is attributed to the interlayer stacking of aromatic segments in the (002) crystal plane (PDF#87-1526). These two diffraction peaks are in good agreement with the previously reported g-C_3_N_4_ [30,31,32]. Notably, P-SQDs/g-C_3_N_4_ shows a diffraction peak different from that of g-C_3_N_4_, a new peak located at 10.6° (d = 0.83 nm). This peak, reported by other articles, is related to the hydrolysis/oxidation of g-C_3_N_4_, which reduces the polymerization of g-C_3_N_4_ and promotes the formation of g-C_3_N_4_ containing oxygen functional groups [31,33,34]. During the hydrothermal treatment, the alignment of the tri-s-triazine units changes due to the hydrolysis/oxidation process. The peak at 27.6° corresponds to the interlayer stacking of the aromatic segments at a distance of 0.32 nm. The XRD peak in Figure 3a shifts from 27.3° for g-C_3_N_4_ to 27.6° for P-SQDs/g-C_3_N_4_, and the interlayer distance also decreased from 0.326 nm to 0.322 nm accordingly. It is also observed that the characteristic peaks of P-SQDs/g-C_3_N_4_ are much more pronounced than those of g-C_3_N_4_, implying a higher degree of π-electron conjugation. In addition, the ratio of 27.6° to 10.6° peaks in P-SQDs/g-C_3_N_4_ is greater than 1, further indicating the formation of higher-order π–π stacking in P-SQDs/g-C_3_N_4_ [35,36,37]. XRD images show that P-SQDs/g-C_3_N_4_ has good crystallinity, while the diffraction peaks of the composite material are enhanced and long-range ordering was improved.

FTIR spectra are shown in Figure 3b. The broad signal of P-SQDs at 1097 cm^−1^ is caused by the stretching vibration of C-O-H or C-O-C [24]. The C-C bending vibration peak (1097 cm^−1^) in P-SQDs/g-C_3_N_4_ composites is not significant compared to P-SQDs, which may be due to the C-C bond breakage in P-SQDs [35]. The absorption peak at 814 cm^−1^, which is present in both pure g-C_3_N_4_ and P-SQDs/g-C_3_N_4_, is attributed to the tri-s-triazine unit [38]. The peaks at 1200–1600 cm^−1^ are the stretching vibrations of the aromatic C-N groups [39], while the 3000~3500 cm^−1^ broad absorption peaks are caused by the stretching vibrations of N-H and -OH [17]. Based on the positions of the characteristic peaks of P-SQDs/g-C_3_N_4_, it can be seen that the addition of P-SQDs did not change the structure of g-C_3_N_4_. Additionally, the intensities of these characteristic peaks in P-SQDs/g-C_3_N_4_ are even higher than those of pure g-C_3_N_4_, indicating that more tri-s-triazine units and aromatic C-N heterocycles were formed while the C-C bonds were broken in the biochar. More tri-s-triazine units and aromatic heterocycles can induce stronger π–π interactions in the P-SQDs/g-C_3_N_4_ composites, which facilitates the electron delocalization effect and further promotes electron transport in the photocatalytic process.

We investigated the optical properties of the synthesized P-SQDs and SQDs (Figure 3c,d). P-SQDs have a maximum emission intensity of 490 nm under a 400 nm excitation, while SQDs show maximum emission intensity of 505 nm at a 420 nm excitation. As the excitation wavelength increased from 300 to 480 nm, the emission wavelength of P-SQDs gradually red-shifted from 445 to 542 nm, indicating the occurrence of photon reabsorption. The emissive intensities are also excitation-dependent, coming up with a decrease–increase–decrease tendency. In addition, Figure 3c,d shows that at the same excitation wavelength (400 nm), the PL of P-SQDs undergoes a significant red shift compared to SQDs. It is mainly due to the increased conjugation of phosphorus elements after doping.

The chemical composition of P-SQDs was further characterized through XPS. As seen in Figure 4a, elements of C, O, S, and P were found in P-SQDs. It can be seen from the spectrum that 284.8, 531.3, 230.9, 165.3, 188.3 and 132.5 eV were attributed to C 1s, O 1s, S 2s, S 2p, P 2s, and P 2p, respectively. In Figure 4b, there are four major peaks at 287.4 eV (C=O), 285 eV (C–P), 284.8 eV (C–C), 283.4 eV (C=C), on the C 1s spectrum of P-SQDs [40]. The S 2p spectra of P-SQDs (Figure 4c) show five sulfur species exhibited in the reaction mixture, including SO_4_^2−^ (168.4 eV), SO_3_^2−^ (167.2 eV), C–S (165.5 eV), S_x_^2−^ (162.4 eV), and divalent sulfur ions (S^2−^ and S_x_^2−^) (161.5 eV), respectively [20,26]. Two peaks at 133.4 and 134.2 eV show the P 2p spectra (Figure 4d), which confirmed the presence of P–C and P–O bonds [41].

### 2.3. Band Structure and Photoelectric Properties

The UV–vis absorption spectra of P-SQDs/g-C_3_N_4_ and g-C_3_N_4_ are shown in Figure 5a. Pure g-C_3_N_4_ has a clear absorption edge at about 450 nm [42]. When P-SQDs are complexed onto g-C_3_N_4_, the absorption band redshifts at about 465 nm, indicating a higher band to band absorption ability of visible light. According to Tauc method, the band gaps of P-SQDs/g-C_3_N_4_ and g-C_3_N_4_ are 2.96 and 3.00 eV, respectively (inset of Figure 5a). At the same time, the absorbance of P-SQDs/g-C_3_N_4_ in the visible range (465 nm to 800 nm) is increased by a factor of 2.78, also representing a better absorption in visible light [43].

To investigate the separation and transfer of photogenerated carriers in P-SQDs/g-C_3_N_4_, we further conducted PL, TPR, and EIS measurements. Figure 5b shows the PL spectra under the excitation wavelength of 345 nm. A strong photocarrier recombination of g-C_3_N_4_ is exhibited with an emissive peak centered at 460 nm. The PL spectra of P-SQDs/g-C_3_N_4_ show a slight blue shift of 25 nm compared to g-C_3_N_4_, which is attributed to the quantum confinement effects [44]. In contrast, the emission intensity of P-SQDs/g-C_3_N_4_ is significantly lower than that of g-C_3_N_4_, which indicates that P-SQDs/g-C_3_N_4_ has a lower direct recombination efficiency of the photogenerated electron-hole (e–h) pairs. The transient photocurrent curves (Figure 5c) reveal that the lower e–h direct recombination ratio comes from the charge separation and transfer. Under the white light of a 300 W xenon lamp, the photocurrent response of P-SQDs/g-C_3_N_4_ is significantly enhanced compared to that of g-C_3_N_4_, indicating a fast and reversible photocurrent response, which means that in P-SQDs/g-C_3_N_4_, more photogenerated carriers can participate in current conduction while not directly recombining to emit PL. Namely, the charge separation and transfer in P-SQDs/g-C_3_N_4_ are more effective. At the same time, EIS analyses were performed and are shown in Figure 5d. P-SQDs/g-C_3_N_4_ exhibits a smaller arc radius, also demonstrating a higher charge separation and transfer efficiency [45]. Combined with the above experimental results, the compounding of P-SQDs with g-C_3_N_4_ leads to an increase in visible light utilization and improves the separation and transfer efficiency of photogenerated carriers, which favors a better photocatalytic performance.

### 2.4. Photocatalytic Performance

The photocatalytic performances of g-C_3_N_4_, P-SQDs, SQDs/g-C_3_N_4_, P-SQDs-1/g-C_3_N_4_, P-SQDs-2/g-C_3_N_4_, and P-SQDs-3/g-C_3_N_4_ were tested with a 300 W Xe lamp, respectively (Figure 6a). Under the same experimental conditions, after 120 min, MO itself hardly degraded under white light, and g-C_3_N_4_ can only photodegrade 63.36% MO, while P-SQDs-2/g-C_3_N_4_ showed excellent photocatalytic degradation efficiency, 98.72%. Notably, SQDs showed no observable photodegradation ability in our experiments. The MO degradation rates of P-SQDs-1/g-C_3_N_4_, P-SQDs-2/g-C_3_N_4_, and P-SQDs-3/g-C_3_N_4_ were 87.35%, 98.72%, and 34.69%, respectively. Too much P-SQDs addition leads to a the decrease in the photocatalytic activity, it may be attributed to the fact that excessive P-SQDs would block the effective light contact area of g-C_3_N_4_. This situation is also observed in graphene quantum dots and carbon nitride composites [17]. The kinetic behavior was described using the first-order kinetic equation [ln(C_t_/C_0_) = kt, where k is the reaction rate constant, C_t_ is the MO concentration at reaction time t, and C_0_ is the initial MO concentration]. The value of k of P-SQDs-2/g-C_3_N_4_ was 3.9 times higher than that of g-C_3_N_4_ (Figure 6b). From the perspective of practical application, the stability and durability of photocatalysts are also important. To confirm the stability and durability of P-SQDs/g-C_3_N_4_, repeated experiments were performed under the same experimental conditions (Figure 6c), and it can be seen that the catalytic efficiency of P-SQDs-2/g-C_3_N_4_ remained unchanged after three repetitions. Figure 6d shows the variation of the MO absorption peak after P-SQDs/g-C_3_N_4_ photodegradation, and the decrease in its absorption ability and the blueshift of the peak indicates that MO was decomposed and its structure also changed.

### 2.5. Analysis of the Photocatalytic Mechanism

In order to have a deeper understanding of the roles of the active components in the photocatalytic process, P-benzoquinone (p-BQ), potassium dichromate (K_2_Cr_2_O_7_), isopropanol (IPA), and potassium iodide (KI) were added as scavengers for the sudden inhibition of superoxide radicals (·O_2_^−^), electrons (e^−^), hydroxyl radicals (·OH), and holes (h^+^), respectively. As shown in Figure 7, without the addition of scavengers, the reaction system was essentially completely degraded, indicating that no free radicals were hindered. While the degradation rate of the reaction system with the addition of p-BQ and KI was inhibited, which indicates that the ·O_2_^−^ and h^+^ radicals in the catalyst play a crucial role in the photocatalytic process. The degradation rate of the reaction system with the addition of K_2_Cr_2_O_7_ was also reduced, indicating that e^−^ is also one of the active substances for the photocatalytic degradation of P-SQDs/g-C_3_N_4_. The effects of the active substances were ranked as ·O_2_^−^ > h^+^ > e^−^.

According to the Mott–Schottky curves in Figure 8a,b, it is known that the conduction band minimum (CBM) of P-SQDs/g-C_3_N_4_ is −0.55 eV, and a bit higher than g-C_3_N_4_ (−0.57 eV). Combined with the band gap analyses (Figure 5a), we give the band structures of these samples. Thus, a possible mechanism for the photocatalytic degradation of P-SQDs/g-C_3_N_4_ was developed. As shown in Figure 8, g-C_3_N_4_ can absorb a part of visible light; therefore, photogenerated electrons are excited from the valence to the conduction band under the visible light. The forementioned TPR and EIS results have shown that P-SQDs can promote the separation of the photogenerated e–h pairs in g-C_3_N_4_ to suppress their direct recombination; thus, O_2_ is converted to ·O_2_^−^ by the photogenerated electrons, while water is decomposed to OH by the holes. MO is oxidized to CO_2_ and H_2_O by the active groups. Moreover, compared to g-C_3_N_4_, the enhancement of visible light absorption due to the narrower band gap of P-SQDs/g-C_3_N_4_ (Figure 5a), will further improve the photocatalytic performance and decompose MO in a shorter time.

## 3. Experimental Section

### 3.1. Material

Sulfur Sublimed (S, ≥99.5%) was purchased from Chengdu Kolon Chemical Co., Ltd. (Chengdu, China). Polyethylene glycol (PEG, Mn = 400 Da), sodium hydroxide (NaOH, ≥96.0%), disodium hydrogen phosphate dodecahydrate (Na_2_HPO_4_·12H_2_O, ≥99.0%), urea (H_2_NCONH_2_, ≥99.0%), and methyl orange (MO) were purchased from Xilong Science Co., Ltd. (Shantou, China). All the raw materials were of analytical grade and used without further purification. The experimental water was deionized.

### 3.2. Preparation of P-SQDs

We prepared P-SQDs using a hydrothermal method. SQDs were synthesized according to a previously reported procedure [25], as follows: 1.6 g S, 3 g PEG, 4 g NaOH, and 100 mL deionized water were stirred at 90 °C for 10 h. After cooling to room temperature, impurities were removed with a 0.22 μm microporous membrane. Next, 1.2 g Na_2_HPO_4_ was dissolved in 30 mL SQDs and treated with ultrasonication (500 W, 40 kHz) for 30 min, and then put into a Teflon autoclave and heated at 180 °C for 10 h. After cooling to room temperature, it was dialyzed, and finally freeze-dried to obtain P-SQDs.

### 3.3. Preparation of P-SQDs/g-C_3_N_4_

The g-C_3_N_4_ material was prepared by a thermal polycondensation method [17]: Putting 25 g urea into a crucible with a lid, and heating it in a muffle furnace at a rate of 2.5 °C/min until 550 °C to hold for 3 h, and then naturally cooling it to room temperature.

The P-SQD/g-C_3_N_4_ photocatalyst was prepared using a simple hydrothermal method. Firstly, 0.6 g g-C_3_N_4_ with X g P-SQD (X = 0.2, 0.4, 0.6) separately was dissolved in 80 mL water, after 30 min sonication, and then put into a Teflon autoclave, heated at 180 °C for 6 h and cooled to room temperature. After centrifugal cleaning for three times and freeze-drying, the final products were obtained and named P-SQD-1/g-C_3_N_4_, P-SQD-2/g-C_3_N_4_, and P-SQD-3/g-C_3_N_4_, respectively. Figure 9 briefly illustrates the experimental procedure of this study.

### 3.4. Characterization

The ultraviolet-visible (UV-vis) absorption spectra were recorded using a PerkinElmer Lambda 750 spectrophotometer (CT, Houston, TX, USA). The Fourier transform infrared (FTIR) spectra were measured with a NICOLET-6700 FTIR spectrometer (Tokyo, Japan). The crystal structure of the samples was studied by MiniFlex-600 (JEOL, Japan) X-ray diffractometer (XRD). Transmission electron microscopy (TEM, JEM-2100F (Tokyo, Japan)), scanning electron microscopy (SEM, KYKY-EM6900) and X-ray energy dispersive spectrometry (EDS, S-00123, USA) were used to study the particle size, morphology and elemental composition of the samples. The specific surface area was calculated by applying the Brunauer–Emmett–Teller (BET, JW-BK112) model to the adsorption data. The valence distribution and bonding patterns of the sample elements were measured by an ESCALAB-250XI (Waltham, MA, USA) X-ray photoelectron spectrometer. Photoluminescence (PL) spectra were measured on a fluorescence spectrophotometer (Edinburgh FL/FS900 Carry Eclipse, Cheadle, UK). All the measurements were conducted at room temperature.

### 3.5. Electrochemical Measurements

The transient photocurrent response spectra (TPR), Mott–Schottky curves and electrochemical impedance spectra (EIS) Nyquist plots were performed with an electrochemical workstation (CHI 760E, Shanghai, China) under a standard three-electrode system (Pt-counter electrode, saturated glyceryl electrode-reference electrode, sample/FTO-working electrode). Na_2_SO_4_ solution (0.5 M) was used as the electrolyte and a 300 W xenon lamp was used as the light source to measure the photocurrent response. The synthesized photocatalyst was deposited on the electrode using the following method: 0.01 g of the sample was dispersed into ethanol to form a suspension, and then, the formed homogeneous suspension was transferred to the FTO glass electrode. Finally, the prepared electrode was dried at 70 °C for 12 h.

### 3.6. Photocatalytic Degradation Experiments

Methyl orange (MO) solution was chosen to simulate the pollutants to study the photocatalytic properties of the samples. The experiments were conducted at room temperature as a whole. Separately, 50 mg of samples were placed into 50 mL of MO solution (10 mg/L) and firstly stirred in a dark environment for 30 min to ensure that the catalyst reached adsorption–desorption equilibrium with the pollutant. Then, the samples were irradiated under a xenon lamp (300 W), and 3 mL of the reaction solution was taken every 30 min and filtered through a 0.22 μm polyethersulfone (PES) needle filter to remove the suspended catalyst. Finally, the MO concentration at 463 nm was measured using UV–vis spectrophotometers.

## 4. Conclusions

In summary, excitation-dependent fluorescent P-SQDs with ultra-small size and narrow size distribution were synthesized for the first time at ambient conditions. Complexes of P-SQDs with g-C_3_N_4_ were obtained using a simple hydrothermal synthesis strategy, and P-SQDs/g-C_3_N_4_ exhibited good and stable photocatalytic effects. Its MO photodegradation rate was as high as 98.72%, much higher than that of pure g-C_3_N_4_, and it also exhibited good cycling stability. An increase in the specific surface area of the photocatalytic material, a decrease in the band gap width, an increase in the separation efficiency of photogenerated electron–hole pairs, and an enhancement and expansion of visible light absorption were achieved, resulting in significant photocatalytic activity and kinetics. We highlight that a novel metal-free quantum dot was developed and it can be used as an efficient photocatalyst. This work not only shows the great potential of P-SQDs/g-C_3_N_4_ as a photocatalyst, but also reveals a multi-layered structure for dramatically enhancing the photocatalytic activity by adding non-metallic material quantum dots.

## Figures and Tables

**Figure 1 molecules-28-03637-f001:**
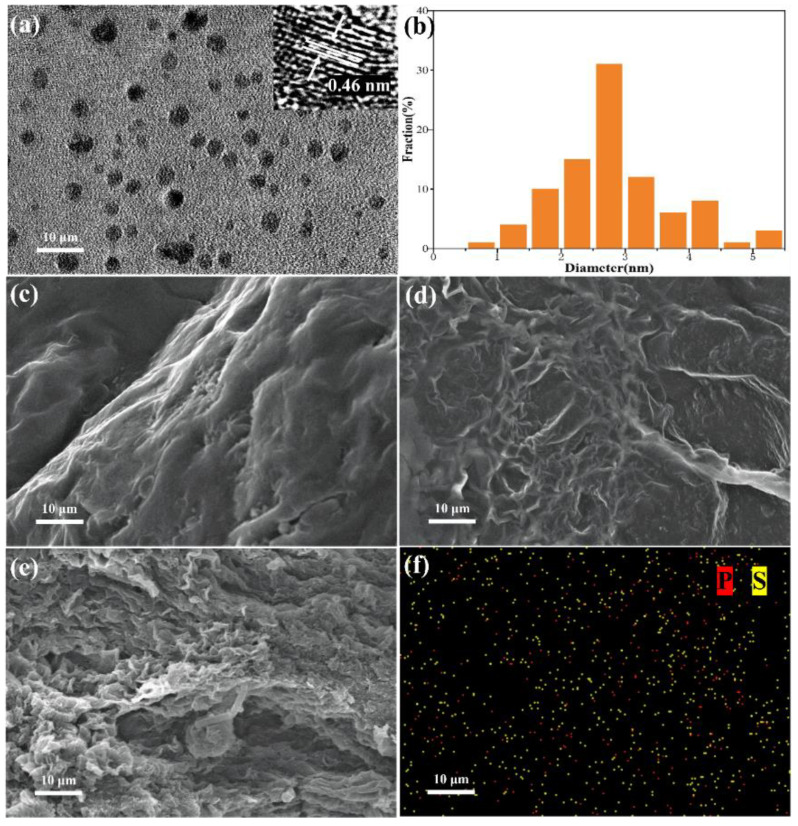
(**a**) TEM image of P-SQDs, the inset is a HRTEM image of P-SQDs showing the lattice spacing, (**b**) particle size distribution of P-SQDs. SEM images of (**c**) g-C_3_N_4_ and (**d**,**e**) P-SQDs/g-C_3_N_4_, (**f**) the EDS elemental map image of P-SQDs/g-C_3_N_4_.

**Figure 2 molecules-28-03637-f002:**
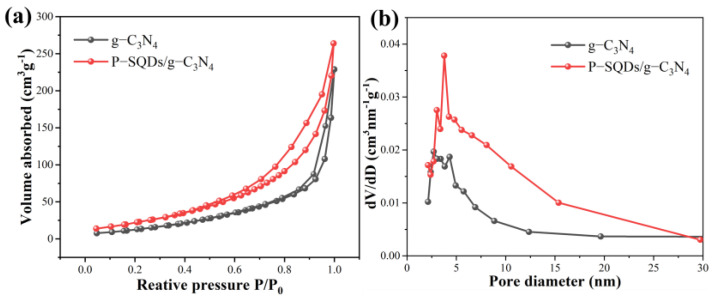
(**a**) N_2_ adsorption–desorption isotherms and (**b**) pore size distribution curves of g-C_3_N_4_ and P-SQDs/g-C_3_N_4_.

**Figure 3 molecules-28-03637-f003:**
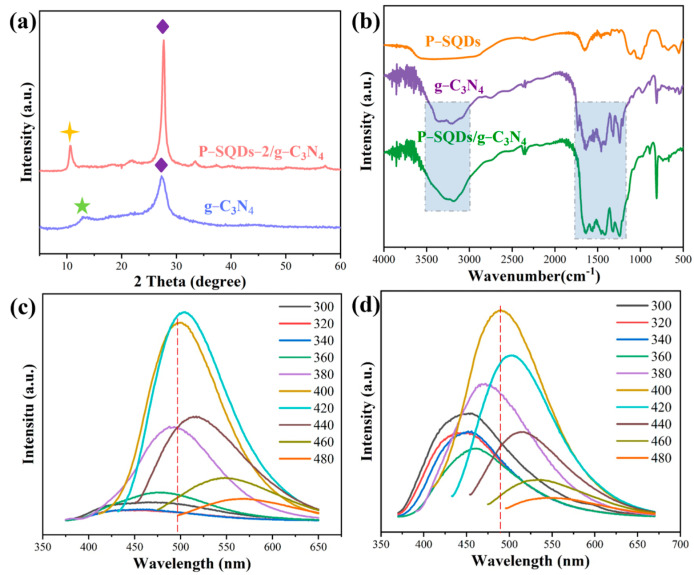
(**a**) XRD patterns of g-C_3_N_4_ and P-SQDs/g-C_3_N_4_. (**b**) FTIR images of P-SQDs, g-C_3_N_4,_ and P-SQDs/g-C_3_N_4_. PL spectra of (**c**) SQDs and (**d**) P-SQDs at different excitation wavelengths.

**Figure 4 molecules-28-03637-f004:**
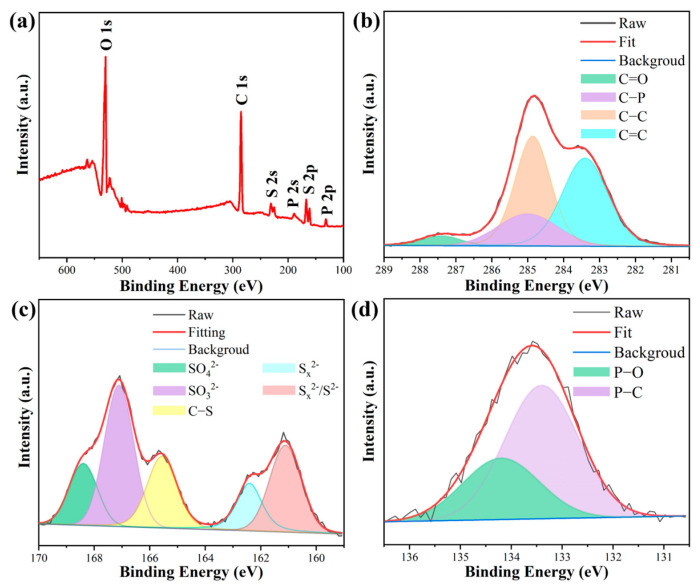
(**a**) XPS survey of P-SQDs. The corresponding high-resolution XPS spectra for (**b**) C 1s, (**c**) S 2p and (**d**) P 2p.

**Figure 5 molecules-28-03637-f005:**
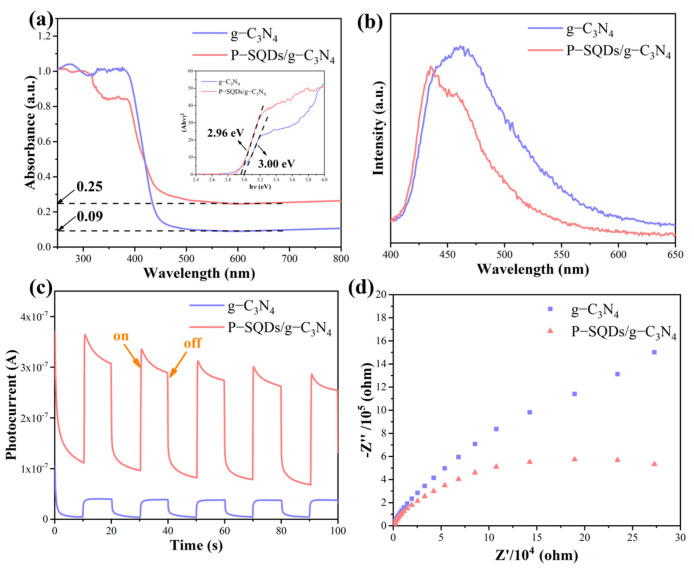
The comparation of P-SQDs/g-C_3_N_4_ and g-C_3_N_4_. (**a**) UV–vis absorption spectra (the insert are band gap simulations using Tauc function). (**b**) Steady-state PL spectra. (**c**) Transient photocurrent curves. (**d**) EIS.

**Figure 6 molecules-28-03637-f006:**
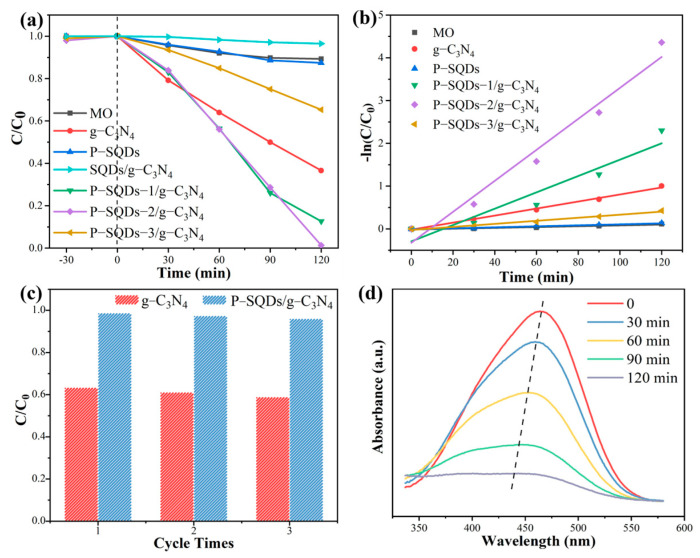
(**a**) Effect of different samples on the photocatalytic degradation of MO under visible light (Xe lamp). (**b**) The corresponding degradation kinetic behavior. (**c**) The photocatalytic cycle tests of g-C_3_N_4_ and P-SQDs/g-C_3_N_4_. (**d**) UV–vis spectra of MO solution after P-SQDs/g-C_3_N_4_ degradation with time.

**Figure 7 molecules-28-03637-f007:**
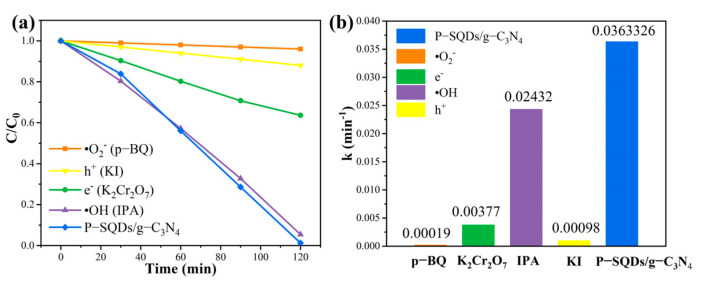
(**a**) Effect of scavengers on the catalytic effect of P-SQD/g-C_3_N_4_. (**b**) The reaction rate of P-SQD/g-C_3_N_4_ in trapping experiments under visible light irradiation.

**Figure 8 molecules-28-03637-f008:**
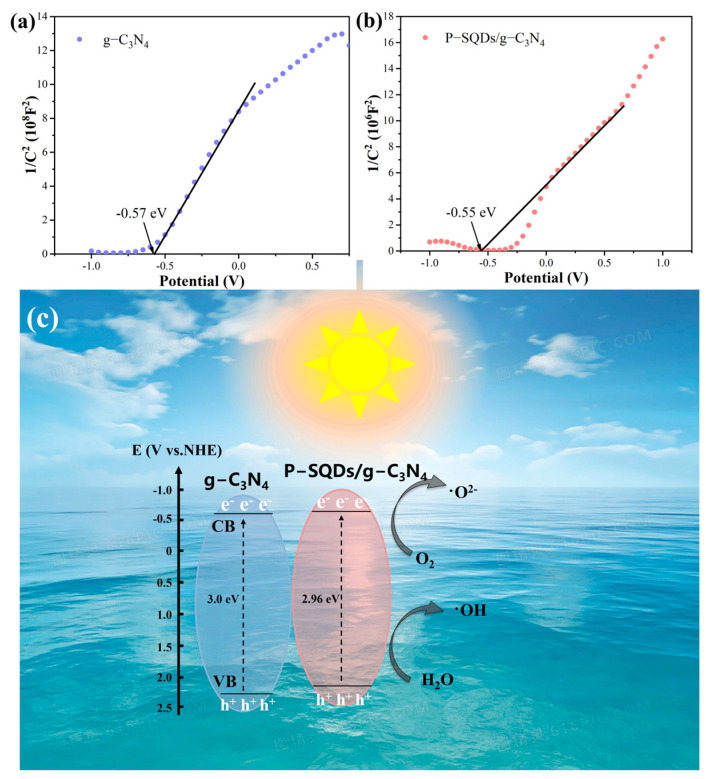
Mott-Schottky curves of (**a**) g-C_3_N_4_ and (**b**) P-SQDs/g-C_3_N_4_. (**c**) Schematic diagram of photocatalytic mechanism of the P-SQDs/g-C_3_N_4_ under visible light irradiation.

**Figure 9 molecules-28-03637-f009:**
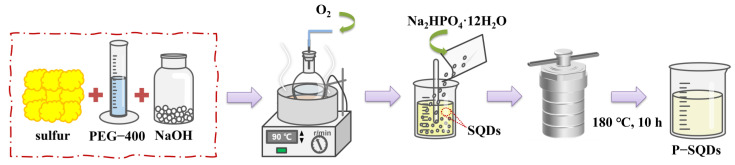
Preparation process scheme of P-SQDs/g-C_3_N_4_.

## Data Availability

The data can be made available upon reasonable request.

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
