# Peer review of "A Novel Non-Metallic Photocatalyst: Phosphorus-Doped Sulfur Quantum Dots"

_molecules, 2023, doi:10.3390/molecules28083637_

Round 1
Reviewer 1 Report
In lines 13 and 14 of the introduction, the authors indicated the superior in easy synthesis and unique electronic band structure and high physicochemical stability of g-C3N4 materials, however, as it is a key part of the job, it should supported with more articles.
Â
Â
Â
the authors should rewrite the first sentence of the second page since they give the idea that there are only 3 articles for the synthesis of SQD, but in the rest of the paragraph they contradict themselves: Currently, only a few works have reported the synthesis of SQDs [15-17].
Put the bibliography in the correct format, see example:Â
Xue, H.; Ren, W.; Denkinger, M.; Schlotzer, W.; Wischmeyer, P.E. Nutritional modulation of cardiotoxicity and anticancer efficacy related to doxorubicin chemotherapy by glutamine and !-3 polyunsaturated fatty acids. J. Parenter Enteral Nutr. 2015, 40, 52–66.
in the first paragraph of page 5, the authors should write the plane assigned to the g-C3N4 catalyst, as well as the JCPD card.
Â
In the 6a graph, the authors should have included the behavior of the catalysts in the 30 minutes of the adsorption-desorption equilibrium or at least the adsorption reaction of the most active material throughout the 120 minutes of reaction, to corroborate the photodegradation activity and discard the idea of MO adsorption
Author Response
In lines 13 and 14 of the introduction, the authors indicated the superior in easy synthesis and unique electronic band structure and high physicochemical stability of g-C3N4 materials, however, as it is a key part of the job, it should supported with more articles.
Response:
--More references have been added to the text (References: 10-14).
Â
the authors should rewrite the first sentence of the second page since they give the idea that there are only 3 articles for the synthesis of SQD, but in the rest of the paragraph they contradict themselves:Â Currently, only a few works have reported the synthesis of SQDs [15-17].
Response:
--This sentence has been rewritten:  Elemental sulfur is one of the most abundant substances on Earth and has been widely used for centuries. In the last few years, SQDs, a new class of non-metallic fluorescent nanomaterials have attracted great interest due to their unique optical, spectroscopic, chemical and antimicrobial properties [21, 22]. In 2014 Li et al. demonstrated for the first time that zero-valent sulfur monomers can be prepared as fluorescent quantum dots and pointed out their potential for energy and photocatalytic applications [23]. At present, great progress has been made in the synthesis of SQDs.
Â
Put the bibliography in the correct format, see example:Â
Xue, H.; Ren, W.; Denkinger, M.; Schlotzer, W.; Wischmeyer, P.E. Nutritional modulation of cardiotoxicity and anticancer efficacy related to doxorubicin chemotherapy by glutamine and !-3 polyunsaturated fatty acids. J. Parenter Enteral Nutr. 2015, 40, 52–66.
Response:
-- All the bibliography formats have been modified.
Â
in the first paragraph of page 5, the authors should write the plane assigned to the g-C3N4 catalyst, as well as the JCPD card.
Response:
-- The JCPD card has been added to the text.
Â
 In the 6a graph, the authors should have included the behavior of the catalysts in the 30 minutes of the adsorption-desorption equilibrium or at least the adsorption reaction of the most active material throughout the 120 minutes of reaction, to corroborate the photodegradation activity and discard the idea of MO adsorption
Response:
-- The behavior of the catalysts of the adsorption-desorption equilibrium in the 30 minutes has been added in Figure 7a.

Reviewer 2 Report
The manuscript submitted by Liu et al. describes the Synthesis and Photocatalysis of Phosphorus-Doped Sulfur Quantum Dots.
The manuscript is presented well, the introduction and the results are written in good way. However, authors should carefully revise the manuscript based on the following comments before it can be considered further for publication.
1- In the title it is very important to mention the word Novel photocatalyst, also in the introduction section, to present the importance of the prepared photocatalysts.
2- The quality of fig 1 is very poor, please maximize it.
3- The BET analysis is missing, and it is very important and mandatory for this manuscript.
4- Did the authors evaluate the effect of concentration for pollutants, PH, amount of the catalysts?
Â
5- please cite the following ref in the introduction section (Magnetic Metal Oxide-Based Photocatalysts with Integrated Silver for Water Treatment) (Magnetic TiO2/CoFe2O4 Photocatalysts for Degradation of Organic Dyes and Pharmaceuticals without Oxidants)
Â
Author Response
The manuscript submitted by Liu et al. describes the Synthesis and Photocatalysis of Phosphorus-Doped Sulfur Quantum Dots.
The manuscript is presented well, the introduction and the results are written in good way. However, authors should carefully revise the manuscript based on the following comments before it can be considered further for publication.
- In the title it is very important to mention the word Novel photocatalyst, also in the introduction section, to present the importance of the prepared photocatalysts.
Response:
-- The title and introduction have been revised.
Â
- The quality of fig 1 is very poor, please maximize it.
Response:
-- Figure 1 has been enlarged.
Â
- The BET analysis is missing, and it is very important and mandatory for this manuscript.
Response:
-- BET analysis has been added to the text: The specific surface area and pore structure of g-C3N4 and P-SQDs/g-C3N4 photocatalysts were investigated by nitrogen adsorption-desorption analysis, and the results are shown in Fig. 3. The specific surface area and pore size of P-SQDs/g-C3N4 composites are superior to those of pure g-C3N4. In Fig. 3a, it can be seen that both g-C3N4 and P-SQDs/g-C3N4 are type IV isotherms with hysteresis loops. The increase in specific surface area of P-SQDs/g-C3N4 is due to the increase in pore volume in the diameter range of 2―30 nm (Fig. 3b), corresponding to the accumulation of broken small particles due to partial polymerization [28]. P-SQDs/g-C3N4 exhibit the largest pore volume in the range of 2―5 nm in diameter [29]. It was demonstrated that by modifying g-C3N4 with P-SQDs, not only the specific surface area can be increased, but also the pore size can be increased to make them richer in active sites, which is conducive to improve the adsorption capacity of the composites on dye wastewater.
Â
Figure 3. (a) N2 adsorption–desorption isotherms and (b) pore size distribution curves of g-C3N4 and P-SQDs/g-C3N4
Â
4- Did the authors evaluate the effect of concentration for pollutants, PH, amount of the catalysts?
 Response:
-- The MO solution of 10 mg/L or 20 mg/L is weakly alkalescent, and after we add a small amount of hydrochloric acid to make it weakly acidic, the color of the MO solution changes, and the characteristic peak is not 463 nm, but red-shifted to 507 nm. Because MO contains many groups, it is reasonable to assume that at this time MO has chemically reacted with hydrochloric acid and is no longer just MO, so we think that the catalytic effect is hard to evaluate when changing the PH of the solution.
Â
-- After adjusting the concentration of MO solution, we can see that when the concentration of MO solution increases, the catalytic effect of catalysts will be significantly weakened.
Â
5- please cite the following ref in the introduction section (Magnetic Metal Oxide-Based Photocatalysts with Integrated Silver for Water Treatment) (Magnetic TiO2/CoFe2O4 Photocatalysts for Degradation of Organic Dyes and Pharmaceuticals without Oxidants)
 Response:
-- These two papers have been added in the Introduction section of the text (References: 7, 8).
Â
Â
